# Current Developments in the Epidemiology and Control of Enzootic Bovine Leukosis as Caused by Bovine Leukemia Virus

**DOI:** 10.3390/pathogens9121058

**Published:** 2020-12-18

**Authors:** Paul C. Bartlett, Vickie J. Ruggiero, Holden C. Hutchinson, Casey J. Droscha, Bo Norby, Kelly R. B. Sporer, Tasia M. Taxis

**Affiliations:** 1College of Veterinary Medicine, Michigan State University, East Lansing, MI 48824, USA; woodsvi1@msu.edu (V.J.R.); norby@msu.edu (B.N.); 2USDA, Fort Collins, CO 80526, USA; holdenhutchinson@gmail.com; 3CentralStar Cooperative, East Lansing, MI 48910, USA; Casey.Droscha@mycentralstar.com (C.J.D.); Kelly.Sporer@mycentralstar.com (K.R.B.S.); 4Department of Animal Science, Michigan State University, East Lansing, MI 48824, USA; taxistas@msu.edu

**Keywords:** epidemiology, dairy, beef, retrovirus

## Abstract

Enzootic Bovine Leukosis (EBL) caused by the bovine leukemia virus (BLV) has been eradicated in over 20 countries. In contrast, the U.S. and many other nations are experiencing increasing prevalence in the absence of efforts to control transmission. Recent studies have shown that BLV infection in dairy cattle has a greater impact beyond the long-recognized lymphoma development that occurs in <5% of infected cattle. Like other retroviruses, BLV appears to cause multiple immune system disruptions, affecting both cellular and humoral immunity, which are likely responsible for increasingly documented associations with decreased dairy production and decreased productive lifespan. Realization of these economic losses has increased interest in controlling BLV using technology that was unavailable decades ago, when many nations eradicated BLV via traditional antibody testing and slaughter methods. This traditional control is not economically feasible for many nations where the average herd antibody prevalence is rapidly approaching 50%. The ELISA screening of cattle with follow-up testing via qPCR for proviral load helps prioritize the most infectious cattle for segregation or culling. The efficacy of this approach has been demonstrated in at least four herds. Breeding cattle for resistance to BLV disease progression also appears to hold promise, and several laboratories are working on BLV vaccines. There are many research priorities for a wide variety of disciplines, especially including the need to investigate the reports linking BLV and human breast cancer.

## 1. Introduction

Enzootic Bovine Leukosis caused by the bovine leukemia virus (BLV) has been eradicated in over 20 countries, while many other nations are experiencing increased prevalence. Recent recognition of BLV’s economic impact has led to increased interest in controlling BLV in our cattle populations. In the last several years, major advancements have improved our understanding of BLV epidemiology, diagnosis and disease control methods, and this review will summarize those advancements and suggest areas for further research.

## 2. Increasing Prevalence

Recent surveys in most nations outside of Europe have reported continuous increases in BLV prevalence in both beef and dairy cattle. Prevalence is generally higher in dairy cattle, with the U.S. Japan, Canada, Brazil, China and Argentina reporting that most herds are infected and are approaching a situation where almost half of their dairy cattle are showing serologic evidence of infection. The following countries and regions around the world are reporting moderate increases in BLV: China [1], Columbia [2,3], Costa Rica [4], Iran [5,6], Japan [7], Mongolia [8], Philippines [9], South Africa [10], Turkey [11] and West Africa [12]. Other nations report higher increases: Argentina [13], Canada [14] and the USA [15,16]. As a result of this increasing prevalence, many areas have implemented BLV-control programs, which include tightened trade restrictions to protect their herds [17].

## 3. Transmission

Most transmission of BLV is through the provirus-infected lymphocyte. Many modes of BLV transmission are possible, but the most important means of transmission likely varies greatly by geography and agricultural system. For example, transmission via fly bites is thought to be important in some parts of the world but not in other parts. Bloodborne transmission is the most recognized route of transmission, but direct contact may be very important in most all management systems. The importance of milk-borne spread is still unknown, as is the potential role of the free RNA-virus. It is unknown if early exposure as a calf or even in utero may lead to a latent infection which later becomes activated, perhaps at times of stress such as pregnancy, first calving and lactation.

## 4. Economic Impact in Dairy Cattle

The economic loss from BLV-induced lymphoma was previously the only known impact of BLV in cattle. Reportedly occurring in <5% of BLV-infected cattle, lymphomatous tumors remain the top cause of condemnation for U.S. dairy cattle carcasses [18]. Like its retrovirus cousin HIV of humans, BLV appears to show most of its impact by facilitating a wide variety of opportunistic pathogens and causes of suboptimal performance. For example, BLV has been associated with infectious diseases such as mastitis, hoof problems and mucosal diseases [19], as well as failure to clear ringworm infection [20,21]. This indirect influence through facilitating other disease conditions probably delayed the cattle industry’s recognition of BLV’s full impact.

BLV-infected dairy cattle may reduce milk production and shorten cow lifespan [18,22]. The impact of BLV infection on milk production has been difficult to measure, since cows which drop in production are usually quickly removed from the herd. Oftentimes, cows that are still present, but designated to be culled, are often excluded from herd production calculations. Some have reported that infected cattle make more milk than BLV-negative cattle [23], or about the same [20,24,25]. However, most researchers have found BLV infection to be associated with lower milk production [1,16,19,26,27,28,29]. In addition, for every 10% increase in BLV prevalence, a loss of 115 kg in rolling herd average was observed [29]. Three large studies (Figure 1) showed a dose-response between rolling herd average milk production and herd BLV prevalence.

One study [29] showing substantial decreased production among older cows showed no decreased production when ELISA-positive animals were compared to their ELISA-negative peers. An analysis of genetic potentials by Wu et al. [30] found that, regardless of blood lymphocyte count, seropositive cattle had a higher genetic potential for milk production than did seronegative herd mates.

BLV-infected cattle were reported to have a shorter lifespan in the herd when compared to their ELISA-negative herd mates [19,23,31,32]. In one analysis, BLV-infected dairy cattle were 23% more likely to be culled than their negative herd mates [31]. A dose-response was seen in cattle with higher ELISA optical density values experienced greater culling. The reasons for the increased culling of BLV-infected cattle are difficult to study because culling decisions are affected by so many issues (such as production, age, disease, market and stocking rates) that are simultaneously and subjectively assessed by the herd managers.

## 5. Altered Immunity

BLV-associated disease and production inefficiencies are thought to be due to a disrupted immune system. A 2015 review by Frie and Coussens [33] outlines several of the altered immune responses reported in BLV-infected cattle. Many of these immune disruptions have clear negative implications for individual and herd health due to infectious disease, especially BLV-associated decreases in systemic immunity (IgG2, IgM), altered activation of immune cells, disrupted T:B cell ratios [21,34,35,36] and diminished response to vaccination [34,37] as indicative of diminished response to naturally presented antigens. BLV infection has also been associated with decreased neutrophil and monocyte function, such as reduced phagocytosis and inhibited apoptosis [38]. Della Libera et al. [39] reported that milk leukocytes (neutrophils and B-lymphocytes) are similarly affected, raising concerns about mammary gland immunity.

The effects of BLV infection on mucosal immunity are not well-characterized despite the importance of mucosal immunity in preventing many cattle diseases. BLV provirus has been detected in a variety of mucosal secretions, including milk, saliva, nasal secretions [40], vaginal secretions and feces [1,41], which raises concerns both for transmission risk and immune function. In a recent report on a small group of cows, average total IgA concentrations were unchanged in serum but were 34% and 24% lower in milk and saliva samples, respectively, in ELISA-positive cows, although neither result reached statistical significance [41]. In contrast, average total IgM concentrations did not differ in milk or saliva but were decreased in serum, as has been observed in other studies [42].

## 6. Proviral Load (PVL)

Multiple copies of the BLV provirus can integrate into the host cell genome at multiple sites [43]. Quantitative PCR (qPCR) tests have recently been developed to determine the PVL which is the number of copies of provirus detected per one of several denominators, such as the microgram of extracted genomic DNA [44]. The PVL measures the actual concentration of provirus in the blood sample rather than the BLV-induced antibodies, which are measured by serological tests such as ELISA [45,46]. Viral load or PVL are used in retroviral diagnosis and management as a measure of infectivity, as they reflect the number of infectious particles per unit of blood. For example, viral load is commonly used for monitoring the risk of transmission for humans infected with the related retrovirus human T-cell lymphotrophic virus 1 (HTLV-1) [47,48]. It is, therefore, consistent that BLV PVL is now being used as a measure of BLV infectivity [49,50].

The correlation between ELISA antibodies and PVL is generally weak, but there is a strong correlation (r = 0.855) between BLV PVL and blood lymphocyte counts [51]. Previous studies have demonstrated that cattle with high lymphocyte counts are more efficient at transmitting BLV [52,53]. Molloy et al. [54] ran experiments in which they reduced the incidence of new BLV infections by selectively culling cows based on viral antigen expression, which they associated with PVL. More recently, Gutiérrez et al. [55] reported increased periparturient transmission of BLV in cows with high PVL, and other investigators have demonstrated that cows with low PVL present a low transmission risk of BLV transmission to their susceptible herd-mates [56,57].

Recently, Hutchinson modeled lymphocyte counts and proviral load over 6-month intervals and measured a small average increase over time [58]. However, this increase was surprisingly small in most cows, and supports the idea that perhaps annual testing is sufficient to identify the most infectious cattle for removal.

## 7. BLV in Beef Cattle

Compared with dairy cattle, much less is known about the epidemiology of BLV in beef cattle. A US Midwest survey completed by Benitez et al. in 2019 showed that 78% of 28 cow-calf herds had at least one ELISA-positive animal, with 36% ELISA-positive cattle among the 3175 tested [59]. A 24-month survival analysis in these herds found that ELISA-positive cattle faced a borderline significant (*P* = 0.06) 18% greater hazard of being culled compared to their ELISA-negative herdmates. In contrast, cattle with a higher proviral load had an 84% greater hazard of leaving the herd as compared with ELISA-negative cattle (*P* = 0.01).

Among breeding beef bulls in the Midwest US, 49% of 39 farms had at least one ELISA-positive bull, and 45% of the 121 bulls tested were ELISA-positive in a recent study [60]. Proviral DNA was identified in the smegma of 7.4% (4/54) of bulls tested in this study, raising the possibility of BLV transmission during natural breeding.

## 8. Controlling BLV

Early control efforts were based on culling cattle with high blood lymphocyte counts [61]. More recently, targeted culling to control BLV has been based on an BLV antibody response. However, eradicating BLV by culling antibody-positive cattle is usually economically impossible for farms with high prevalence rates. Management interventions such as single-use hypodermic needles and examination gloves have not always been effective in reducing transmission [62]. Alternative control approaches are being sought such as vaccination and host genetic selection. Recent development of qPCR tests to measure BLV proviral load are showing efficacy in identifying the most infectious cattle so they can be removed via segregating or culling. Researchers were able to demonstrate in three small herds that selectively segregating or culling of cows with high PVL and high blood lymphocyte counts led to a dramatic decrease in BLV incidence and prevalence within a few years (Figure 2) [63]. Output from the testing laboratory (Central Star Cooperative, E. Lansing, MI, USA) now shows how much each cow contributes to the herd’s total PVL. For example, the manager of one particular farm can see that the 28 cows with the highest PVL accounted for 43% of the herd’s total PVL. This output is proving useful to producers in identifying the most infectious cows to prioritize for removal in order to reduce the incidence of new infections.

One 3000-cow herd in the Midwest U.S. has had great success in screening cows with the ELISA test and measuring PVL on ELISA-positive cows. Cows with the highest PVL were then segregated and prioritized for culling. As of 27 October 2020, only 10 cows with a PVL > 0 remained in a separate pen awaiting culling. Because BLV in the heifers had been controlled via ELISA testing, this large herd is on the verge of eradicating BLV [64].

## 9. Host Genetics

Genetic diversity within a population is directly linked to that population’s ability to respond to specific pathogens, reflecting the high degree of heterogeneity within genes involved in the innate and adaptive immune system. The major histocompatibility complex (MHC) is a cluster of genes that are responsible for self and non-self-antigen recognition for proper immune function in mammals. While the general structure of the MHC is conserved amongst species, specific exons encoding MHC molecules are highly polymorphic to provide a diversity of adaptive immune responses to emerging pathogens within a population. The Bovine leukocyte antigen (BoLA) in cattle is orthologous to the human MHC or Human Leukocyte Antigen (HLA) complex. Within this cluster of evolutionarily conserved genes, the BoLA Class IIa subregion houses the DR and DQ genes, which have undergone extensive genetic characterization within the context of BLV susceptibility and disease progression [65,66,67,68].

Specifically, DRB3 and DQA1 sequence-based typing (SBT) is becoming a requisite component of BLV cohort studies in order to better understand the propensity of individual disease trajectory. The DRB3 gene has 130 identified alleles to date [69] and is the only functional locus within the DR gene; thus, it is the strongest expressed gene within this cluster. In contrast, there are numerous functional DQ genes, including five DQA genes, DQA1 being the most polymorphic with 31 different identified alleles. The second exon of the II BoLA-DRB3 and DQA1 alleles encodes the extracellular portion of the molecule and is polymorphic in nature, which provides antigen-presenting cells with variability in the immune response to particular pathogens for individuals.

Pedigree analysis first demonstrated that bovine lymphosarcoma and persistent lymphocytosis track with family lineages, providing the first evidence for genetic influence on susceptibility to BLV [70]. Originally, serologically defined BoLA Class I alleles were found to be associated with BLV-resistance phenotype [71]; however, MHC class II alleles were later identified as playing a stronger role in BLV outcome in cattle [72]. More recently, SBT of BoLA alleles has found specific BoLA Class II DRB3 and DQA1 alleles to be associated with resistance or susceptibility to the development of BLV-associated disease. Over the last 30 years, a substantial amount of evidence has demonstrated that BoLA class II allele associates with BLV PVL. Although numerous alleles have been classified as leading to a resistant or susceptible BLV phenotype, DRB3*0902 has been most notably associated with the resistant phenotype [65,66,68,72,73,74]. More recently, polymorphism within the DRB3 gene has been identified as having a stronger association with the BLV phenotype than the DQA1 gene [75].

Initial approaches to BoLA typing involved restriction enzyme digestion and analysis of resulting fragments to identify alleles (RFLP). BoLA-allele identification is now accomplished via different sequencing approaches such as PCR followed by Sanger sequencing. Presently, research groups are utilizing multiplex barcoded primers in order to systematically tag hundreds of animals prior to next generation sequencing and bioinformatic analysis, enabling a wider scope of animals. This approach will enable studies to analyze a much larger number of animals to determine how various BoLA-DRB3 alleles associate with disease state. Due to the high degree of heterogeneity of this genetic locus, SNP-based genotyping is unable to resolve the identity via hybridization probes. This genetic analysis would allow bull studs to determine the BoLA status of their young sires and potentially market BLV-resistance as an economically important trait. Use of this approach as an in-herd selection tool may not be as useful, as cattle do not persist within modern commercial dairy herds long enough to recover an economic benefit. Current studies are under way to determine if colostrum from BoLA-DRB3*0902 dams provide additional protection to their young over non-BoLA-DRB3*0902 dams, which may be a useful management strategy for producers. Though selection for this important gene may not resolve the high rate of BLV prevalence in North America, it may be a useful tool for understanding virus-host interaction and BLV’s impact on profitability.

## 10. The BLV Viral Genome

BLV is a retrovirus closely related to HTLV-1, whereby definition, it encodes a reverse-transcribed DNA copy of its RNA genome into the host cell that it invades. Once integrated, the proviral genome is propagated via mitosis. Expression of BLV proviral genes occurs through endogenous transcription and translation mechanisms. The BLV genome consists of a common 5′–gag–pro–pol–env–3′ gene cassette encoding the viral capsid, protease and envelope proteins required for viral repackaging. Additionally, the BLV viral genome encodes its own reverse transcriptase (pol) required for creating the DNA version of the virus for integration accomplished via the virally encoded integrase also located within the pol gene. Accessory genes, including the U3 region within the 3’ untranslated region (UTR) and U5 within the 5’ UTR, enhance viral gene transcription via transcription factor binding sites. Two protein coding genes, Tax and Rex, are considered CIS-transcriptional activators that bind to the 3’ LTR and are commonly implicated in viral oncogenesis [76,77]. Unlike other retroviruses, BLV also encodes 10 microRNAs and other long non-coding RNAs, in which their expression levels are now being shown to associate with BLV pathogenesis [78,79]. Though the genomic architecture is not unique to other retroviruses, such as HTLV-1 and HIV, many knowledge gaps still remain as to how genetic variation within accessory genes may confer differences in transmissibility, virulence and pathogenicity.

## 11. Human Health

The zoonotic potential of BLV has long been debated, but the understanding of retrovirus biology developed for the AIDS/HIV epidemic has led to more recent findings for the BLV retrovirus [80]. Approximately 70% of humans have been reported to have anti-BLV antibodies [81,82], and 25% have detectable BLV provirus in their blood [82,83]. These studies are in contrast to those conducted in the 1970s, which failed to detect BLV antibodies in people with high cattle exposure, such as farmers, veterinarians, meat inspectors and creamery workers [84].

Evidence for human infection has not been reported from any investigator. Whole genome sequencing of 51 breast cancer tumors showed no evidence of BLV DNA [85]. A query of the Cancer Genome atlas failed to identify BLV proviral DNA among 750 breast carcinomas [86] or BLV transcripts among 810 breast cancer samples [87]. However, case-control studies have reported finding BLV DNA in cancerous breast tissue at higher rates than in benign, non-cancerous tissue [88,89,90]. The relationship between BLV and breast cancer was reported to have an attributable risk of between 37–52% [88,91].

Potential routes of transmission between cattle and people are numerous, including direct contact with infected animals and the consumption of unpasteurized or undercooked dairy and beef products, which have been shown to contain the BLV provirus [92]. The detection of the DNA provirus is more suggestive of an actual infection than are anti-BLV antibodies, which may only indicate exposure and not necessarily infection. Clearly, more work is needed to determine all possible human health effects of BLV exposure. It is worth noting that northern European countries that eradicated BLV many years ago are currently experiencing somewhat higher rates of mammary cancer than in the U.S. A more careful ecological study should be conducted once the European women born post-eradication reach adulthood.

## 12. Vaccines

Vaccination of cattle for BLV is still being investigated. Unlike some other retroviruses, such as HIV, BLV has a relatively stable genome, and thus, an efficacious vaccine may be possible. However, many previous attempts at developing a vaccine for BLV have been unsuccessful [93]. Several research groups are currently working on BLV vaccines. One group has developed an attenuated vaccine by targeted mutations and deletions and is seeking regulatory approval for use [93].

Genetically modified vaccine candidates must contend with additional regulatory requirements. Hopefully any developed vaccine will enable vaccinates to be distinguished from cattle with natural infection, and thus, enhance rather than deter any ongoing BLV control efforts.

## 13. Issues for Future Inquiry

What causes a minority of cattle to develop extremely high proviral load and to what extent is this determined by their genetics or the time at which they become infected (in utero or very early in life via colostrum)?How important are host genetics, whereby certain alleles predispose an animal to a high or low proviral load, antibody titer and/or severity of pathology?Why do first lactation animals seemingly avoid the milk production loss and increased risk of culling that are associated with older ELISA-positive cows?The severe inbreeding of Holstein dairy cattle is such that the effective population size is <50 animals, and they would be considered critically endangered if they were a wild animal species [94]. This raises the question of how long a provirus-infected lymphocyte could survive and propagate upon being introduced into an almost genetically identical cow compared to a genetically distinct cow.Attribution studies are needed to determine the percentage of new BLV infections that come from direct contact, biting fly transmission, shared hypodermic needles, etc.The role of free BLV RNA in disease transmission should be definitively established.Controlled studies are required to measure the extent to which BLV infection contributes to the causation of the major dairy diseases such as mastitis, lameness, infertility, etc.Longitudinal studies should determine how commonly animals are infected with BLV as calves, but the virus becomes sequestered until the stress of birth and lactation. Genetic methods should be developed to determine if newly recognized infections in cows are usually caused by the same strain found in their dams over two years ago, or the same strains found in their contemporaries.The role of BLV in the induction and pathogenesis of human breast cancer and other human diseases should be definitively established.What role does microRNA play in the pathogenesis of BLV, and can it be used for diagnosis or prognosis?A safe and efficacious BLV vaccine should be developed that allows differentiation between vaccinates and natural infection.The likely return on investment for BLV control and eradication should be determined for test and removal using ELISA and by using ELISA screening with proviral load follow-up testing.

## 14. BLV and the Future of Animal Agriculture

BLV control measures must be targeted to the needs of the future, rather than the needs of the present. A recent Rethinkx analysis of animal agriculture predicts a rapid contraction in the dairy and beef industries as alternative methods of producing protein are developed, e.g., fermentation of specific proteins and tissue culture [95]. They predict that demand for cow products will have fallen by 70% by 2030. If the Rethinkx predictions are even close to accurate, then efforts to control BLV would be occurring in a market of rapidly declining numbers of herds and cows. The future cattle industry would need to survive competition with the molecular-food industry, which presumably avoids animal welfare concerns, excessive water and land use, serious environmental contamination with manure or generation of excessive greenhouse gases. Supposedly antibiotic residues can be prevented in the food product and in any waste material. Contamination with our common foodborne pathogens such as *E. coli, Campylobacter* and *Salmonella* will become an issue unique to animal agriculture, as it will be much easier to produce a pathogen-free food product coming out of a controlled fermentation or tissue culture process than it will be to produce such a product from a pathogen-abundant farm environment. As the public’s immune systems become increasingly naïve to our common foodborne pathogens, progressively lower pathogen doses will be capable of causing clinical disease.

To compete with these new food products, the surviving animal industry will need to have adequately solved the previously mentioned issues. Future consumers may not want animal products from herds infected with BLV, which may become recognized as a possible public health hazard and also a cause of animal suffering. If the predicted contraction of herd and cow numbers becomes a reality, many producers will likely forgo controlling BLV if they do not expect their herd to survive for many more years. However, herd managers wanting to reduce the size of their herd may take this as an opportunity for selectively culling BLV-infected cows in the belief that BLV-free herds will be more profitable and appealing to consumers. However, in these uncertain times for the animal industries, many producers may not want to invest funds to eradicate BLV. Some may even decide to wait in the hope that a vaccine is developed that will serve as a useful tool for BLV eradication.

## 15. Conclusions

Heightened BLV research activity has been driven by BLV’s increasing prevalence in many nations and recent recognition of its impact on milk production, cow lifespan and possible public health implications. The means by which BLV disrupts the cow’s immune system are being described in greater detail than was possible before. It must be determined how newly available BLV diagnostic tools can best be used to control this infection. Many important research questions must be addressed regarding BLV epidemiology, immunology and control. It is likely that the future of the animal industries will see a continued effort to control BLV.

## Figures and Tables

**Figure 1 pathogens-09-01058-f001:**
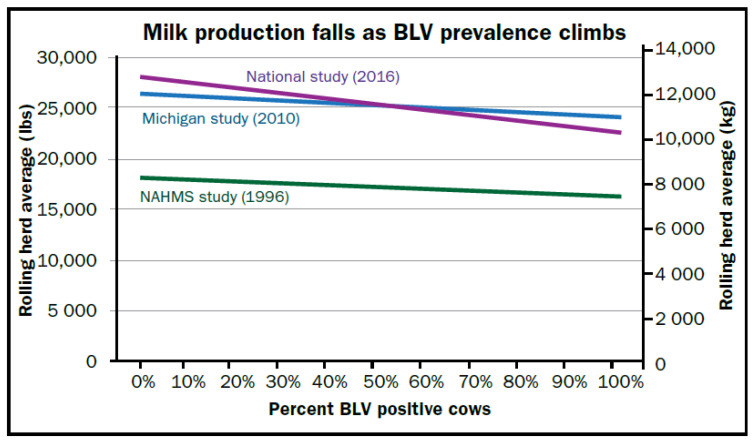
Milk production rolling herd average vs the prevalence of bovine leukemia virus (BLV) ELISA-antibodies from three US studies: LaDronka [16], Erskine [29] and USDA [15].

**Figure 2 pathogens-09-01058-f002:**
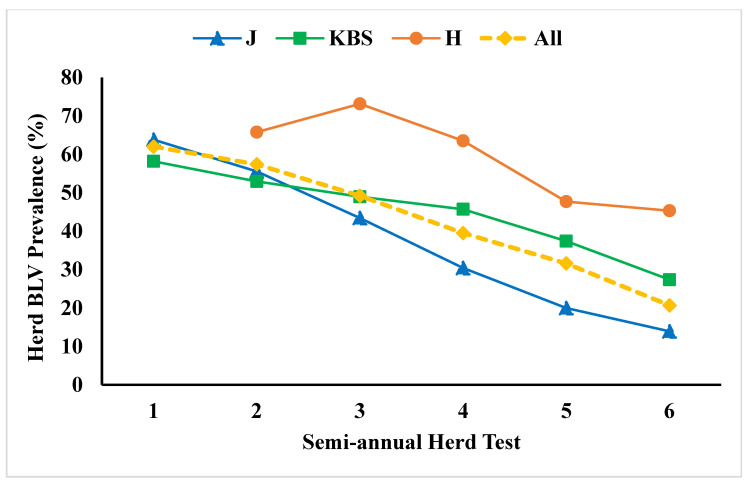
Decrease in herd BLV prevalence among three herds, culling or segregating those cows with the highest lymphocyte counts and/or BLV proviral loads, from Ruggiero [63].

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
