# Peer review of "Current Developments in the Epidemiology and Control of Enzootic Bovine Leukosis as Caused by Bovine Leukemia Virus"

_pathogens, 2020, doi:10.3390/pathogens9121058_

Round 1

Reviewer 1 Report

In this manuscript, Bartlett et al propose a manuscript on different aspects of Bovine Leukemia Virus (epidemiology, immunology, host genetics, economy, control measures, human health, vaccines). Overall, the review is well written. Authors know the field quite well and discuss topics with maturity.

Suggestions for improvement:

- The title only indicates "epidemiology" while the review also (perhaps mainly) presents options to reduce prevalence/incidence.

- Is this paper a review or an article that highlights the interest of segregation or culling based on qPCR for proviral load ? In particular, data from Central Star are presented on lines 162-166.

- Cite and describe the pioneering work of Bendixen (Bendixen HJ, 1960a. Leucosis in cattle in Denmark. II. Pathogenesis and endemic status. Deutsche Tierarztliche Wochenschrift, 67, 57-63. Bendixen HJ, 1960b. Leucosis in cattle in Denmark. III. Clinical picture of transmissible epidemic and sporadic forms. Deutsche Tierarztliche Wochenschrift, 67, 169-173.) BLV was indeed eradicated from Denmark based on lymphocyte counts before the virus was identified.

- Although genetic factors are likely involved in the viral-host interplay, breeding cattle for resistance to BLV always failed despite a number of attempts. A comment/opinion from the authors would be appreciated.

- Several research groups are currently working on BLV vaccines. There is now an efficient vaccine (Abdala 2019 Retrovirology 16: 26).

- Line 77: delete question mark (milk?) 

- Line 78: the reference is extremely old and costs are therefore underestimated. Use only updated costs.

- Line 121: Delete bracket after [44]

- Line 262 and 59. What is the basis of this question? Is there evidence that full length genomic RNA is stable in secretions ?

- Line 216-238. The opinion of the authors about the controversy would be appreciated.

- Line 265. What approach would tackle this question?

- Line 270. What approach would definitely close the controversy ?

- Line 276: Also include the role of infected bull breeders in transmission

- Line 287-305: This paragraph lacks citations

- Line 287 “it is claimed” where ?

Minor comments:

- Line 16: Enzootic Bovine Leukosis is abbreviated EBL (but not BLV).

- Line 52: add that iatrogenic procedures are main routes of transmission. Vets are responsible... because use of infected needles shared for all animals in a herd.

- line 121-125: "commonly used" is used 3 times.

- line 183: Correct "... orthologous to?... ".

- lines 198 and 205 "... associated with BLV phenotype... " is unclear.

- line 202 "...BoLA class II allele associates with BLV PVL... " is unclear.

- lines 206-213. The description of the old RFLP method is unnecessary excess of details. The paragraph should be ended by the opinion of the author about the possibility to use genetic selection based on BoLA alleles (advantages and disadvantages, risks). A comment on the inefficiency of this approach should also be included (as outlined in issue #4 line 255).

- Line 217 "... Retrovirus technology developed for the AIDS/HIV epidemic has led to more recent findings for the BLV retrovirus..." The term "technology" is inadequate.

- Line 253 is unclear

- Line 272. There are 10 microRNAs.

- There are errors in references (lines 437, 487, 504, 581). Some articles are missing doi

Author Response

Response to Reviewers – December 15, 2020

Thank you excellent reviews that helped improve this paper.

Reviewer 1

In this manuscript, Bartlett et al propose a manuscript on different aspects of Bovine Leukemia Virus (epidemiology, immunology, host genetics, economy, control measures, human health, vaccines). Overall, the review is well written. Authors know the field quite well and discuss topics with maturity.

Suggestions for improvement:

- The title only indicates "epidemiology" while the review also (perhaps mainly) presents options to reduce prevalence/incidence.  Good point.  The word “Control” was added to the title

- Is this paper a review or an article that highlights the interest of segregation or culling based on qPCR for proviral load ? In particular, data from Central Star are presented on lines 162-166.  We are attempting to highlight recent developments, and using PVL to target cattle for segregation or culling is one of the major developments.

 - Cite and describe the pioneering work of Bendixen  Done

Bendixen HJ, 1960. Leucosis in cattle in Denmark. II. Pathogenesis and endemic status. Deutsche Tierarztliche Wochenschrift, 67, 57-63.

Bendixen HJ, 1960. Leucosis in cattle in Denmark. III. Clinical picture of transmissible epidemic and sporadic forms. Deutsche Tierarztliche Wochenschrift, 67, 169-173.

BLV was indeed eradicated from Denmark based on lymphocyte counts before the virus was identified. Done

 Although genetic factors are likely involved in the viral-host interplay, breeding cattle for resistance to BLV always failed despite a number of attempts. A comment/opinion from the authors would be appreciated.  These previous attempts were made before it was known which specific alleles conveyed resistance or susceptibility.  Identification of the relevant alleles is ongoing and attempts to use this knowledge for breeding programs is just getting started.  We think the success of this approach is yet to be determined.

- Several research groups are currently working on BLV vaccines. There is now an efficient vaccine (Abdala 2019 Retrovirology 16: 26).  Done

 - Line 77: delete question mark (milk?)  Done

- Line 78: the reference is extremely old and costs are therefore underestimated. Use only updated costs.  We removed this line citing the Ott study.

 - Line 121: Delete bracket after [44]  Done

 - Line 262 and 59. What is the basis of this question? Is there evidence that full length genomic RNA is stable in secretions ?  The basis is that viral transmission occurs with other retroviruses but is not generally thought to play an important role in BLV. 

 - Line 216-238. The opinion of the authors about the controversy would be appreciated.  This is not our area of research so we would rather this review paper just summarize what others have found.

 - Line 265. What approach would tackle this question?  We added that longitudinal studies will be needed, and we are starting one this spring.

 - Line 270. What approach would definitely close the controversy ?  This will take lots of different study designs.  We didn’t go into this topic in great depth but hoped to encourage others in this direction.

 - Line 276: Also include the role of infected bull breeders in transmission  Oscar Benitez’s referenced paper goes into this topic in depth.  Observational studies show that natural breeding is associated with more BLV infection.

 - Line 287-305: This paragraph lacks citations.  These are our original thoughts not previous published.

- Line 287 “it is claimed” where ?  The Rethinkx article makes this claim, but we changed the wording to reflect the fact that we don’t know if they will be successful.

  Minor comments:

 - Line 16: Enzootic Bovine Leukosis is abbreviated EBL (but not BLV).  Done

 - Line 52: add that iatrogenic procedures are main routes of transmission. Vets are responsible... because use of infected needles shared for all animals in a herd. Ten years ago I would have agreed, but lately its looking like direct transmission may be more important.  Stopping shared gloves/sleeves and shared hypodermic needles alone does not seem to stop transmission. And the Japanese are convinced that biting flies are the most important route of transmission in their country. 

 - line 121-125: "commonly used" is used 3 times.  Done.  Thanks for noticing this.

 - line 183: Correct "... orthologous to?... ".  Done

 - lines 198 (now 197) and 205 (now 204) "... associated with BLV phenotype... " is unclear.  Changed to BLV-resistance phenotype

 - line 202 "...BoLA class II allele associates with BLV PVL... " is unclear.

 - lines 206-213. The description of the old RFLP method is unnecessary excess of details. The paragraph should be ended by the opinion of the author about the possibility to use genetic selection based on BoLA alleles (advantages and disadvantages, risks). A comment on the inefficiency of this approach should also be included (as outlined in issue #4 line 255).  This paragraph (now starting on line 205) has been re-written.

 - Line 217 "... Retrovirus technology developed for the AIDS/HIV epidemic has led to more recent findings for the BLV retrovirus..." The term "technology" is inadequate. We changed it to “understanding of retrovirus biology”

 - Line 253 is unclear  Done

 - Line 272. There are 10 microRNAs. Done

 - There are errors in references (lines 437, 487, 504, 581). Some articles are missing doi; references updated with corrected information. DOI added where available.  Done

Reviewer 2 Report

This is a well-written and thoughtful review which provides a comprehensive assessment of the role of bovine leukemia virus (BLV) infections in the U.S. and world cattle industry, with a particular emphasis on its economic impact, culling practices, and the need for improved screening and detection by ELISAs and qPCR. There were only a few minor suggestions for improvement and these are summarized below:

1) It would be helpful if the study included a brief discussion of the BLV viral genome and its products, in particular its regulation by miRNAs, with a comparison to the related human T-cell leukemia virus type-1 and its products.

2) The purported ability of BLV to directly infect humans and the suggestion that it may have a role in human disease are dubious and not well-supported. Perhaps an expanded discussion here of the cellular tropism of BLV and its receptor usage with respect to its ability to cause B-cell lymphomas in cattle and sheep would add to this section.

3) It would also be useful to discuss the various (and numerous) studies using BLV as a model for HTLV-1-induced disease, as these seem to have been overlooked. This would make the review more comprehensive and enhance its broader relevance. Overall, this was a very nice review and will advance our understanding of the epidemiological and economic impact of this zoonotic pathogen.    

Author Response

Response to Reviewers – December 15, 2020

Thank you excellent reviews that helped improve this paper.

Submission Date

19 November 2020

Date of this review

Reviewer 2:

This is a well-written and thoughtful review which provides a comprehensive assessment of the role of bovine leukemia virus (BLV) infections in the U.S. and world cattle industry, with a particular emphasis on its economic impact, culling practices, and the need for improved screening and detection by ELISAs and qPCR. There were only a few minor suggestions for improvement and these are summarized below:

1) It would be helpful if the study included a brief discussion of the BLV viral genome and its products, in particular its regulation by miRNAs, with a comparison to the related human T-cell leukemia virus type-1 and its products. 

We added a paragraph about this

2) The purported ability of BLV to directly infect humans and the suggestion that it may have a role in human disease are dubious and not well-supported. Perhaps an expanded discussion here of the cellular tropism of BLV and its receptor usage with respect to its ability to cause B-cell lymphomas in cattle and sheep would add to this section. 

We added text about this

3) It would also be useful to discuss the various (and numerous) studies using BLV as a model for HTLV-1-induced disease, as these seem to have been overlooked. This would make the review more comprehensive and enhance its broader relevance. Overall, this was a very nice review and will advance our understanding of the epidemiological and economic impact of this zoonotic pathogen. 

We decided to limit this paper to BLV and not discuss HTLV-1.